# SDR-Based 28 GHz mmWave Channel Modeling of Railway Marshaling Yard

**DOI:** 10.3390/s23198108

**Published:** 2023-09-27

**Authors:** Yiqun Liang, Hui Li, Yuan Tian, Yi Li, Wenhua Wang

**Affiliations:** 1Postgraduate Department, China Academy of Railway Sciences, Beijing 100081, China; liangyiqun@139.com; 2Signal and Communication Research Institute, China Academy of Railway Sciences Corporation Limited, Beijing 100081, China; ty1024@rails.cn (Y.T.);; 3National Research Center of Railway Intelligence Transportation System Engineering Technology, Beijing 100081, China; 4National Railway Track Test Center, Beijing 100015, China; wangwenhua02@163.com

**Keywords:** mmWave, 28 GHz, propagation, railway, hotspot, marshaling yard, SDR, channel modeling, obstruction

## Abstract

Compared with railway communication service requirements on the mainline, requirements in hotspots such as stations and yards are more complicated in terms of service types as well as bandwidth, of which railway-dedicated mobile communication systems such as 5G-R facilitated with dedicated frequency support cannot meet the entire communication requirements. Therefore, other radio-communication technologies need to be adopted as a supplement, among which the mmWave communication system is a promising technology, especially for large bandwidth communication between train and trackside. However, there is a lack of evaluation of the 28 GHz mmWave channel characteristics for the railway marshaling yard scenario. In this paper, the railway marshaling yard mmWave propagation scenario is deeply analyzed and classified into three typical categories, based on which, a measurement campaign is conducted using an SDR channel sounding system equipped with a 28 GHz mmWave phased-array antenna. A self-developed software under the LabVIEW platform is used to derive the channel parameters. Conclusions on the relationship between the parameters of MPC numbers, time-spread, and received power and position, as well as the impact of typical obstructions such as the Catenary, adjacent locomotives, and buildings are drawn. The statistical results and conclusions of this paper are helpful for facilitating the design and performance evaluation of future mmWave communication systems for railway marshaling yards and can also be further extended and applied to the research of mmWave utilization in 6G and other future communication technologies for more scenarios.

## 1. Introduction

Railway transportation, as an important pillar of national economics, is directly related to the safety of people’s lives and property and is hailed as the “artery of the country”. As a common consensus, safety is the lifeline of railway transportation. Railway radio communication plays a very important role in railway transportation with respect to transportation production as well as safety management; as a means of communication and information technology, railway radio communication reliably carries the dispatching communication service, train control information, etc., and it is the foundation of promoting production efficiency and the high-quality development of railways.

Radio frequency is the foundation and prerequisite for the innovation and development of railway radio-communication technology. In view of the high reliability requirements of railway radio-communication systems, it is widely recommended by the International Union of Railways (UIC) and other countries in the world to allocate dedicated frequency resources for railway radio-communication systems. The International Telecommunication Union (ITU), UIC, and other countries have paid close attention to the coordination of railway-dedicated frequencies and frequency protection. In 2015, the World Radio-communication Conference (WRC-15) decided to set up agenda item 1.11 to study spectrum harmonization for railway radio-communication systems between train and trackside within the existing mobile service allocations, and countries conducted in-depth research on railway radio-communication systems at the ITU level. In UIC, under the umbrella of the future railway mobile communication system (FRMCS) project, the Frequency Working Group (UGFA) was established and is now responsible for studying the related topics of frequency with respect to railway radio-communication systems.

The dedicated frequency for railway radio communication has played a significant role in railway transportation production, ensuring safety and improving efficiency, as well as achieving huge economic and social benefits. The European Union has allocated 876–880 MHz (train to ground) and 921–925 MHz (ground to train) as dedicated operating frequency bands for GSM-R in Europe, with over 100,000 km of lines currently deployed. In China, GSM-R uses 885–889 MHz (train to ground) and 930–934 MHz (train to ground), and covers over 85,000 km of railway lines.

With the trend of digitalization and intelligence, the contradiction between the increasing service demand and the shortage of frequency resources as well as the limited carrying capacity of railway radio-communication systems has become increasingly prominent. Facing the rapid evolution of radio-communication technology, the broadband of railway radio-communication systems has become a future development direction. Considering the limited frequency resources in the low-frequency band and fragmented allocation, it is an inevitable trend to migrate to the middle- and high-frequency band. In Europe, the existing 2 × 4 MHz R-GSM frequency resources of GSM-R are expanded to the 2 × 5.6 MHz (874.4–880/919.4–925 MHz) ER-GSM frequency band; in addition, 1900–1910 MHz is used as a supplementary frequency band for FRMCS(5G-R). In the meantime, China Railway has submitted an application to the Ministry of Industry and Information Technology(MIIT) for using the 2100 MHz 2 × 10 MHz frequency band as a 5G-R system field test frequency.

However, in addition to meeting the requirements of services on the main lines, such as dispatching communication, train control, monitoring, etc., service models in hotspots such as stations and yards are more complicated in terms of service types as well as bandwidth requirements. The 5G-R system facilitated with dedicated frequency support cannot meet the entire communication requirements of hotspots, so other radio-communication technologies and methods need to be adopted as a supplement to the 5G-R communication system. Among these, the mmWave communication system is a promising technology, especially for large bandwidth communication between train and trackside.

There are fundamental differences between mmWave communications and existing communication systems, in terms of high propagation loss, directivity, and sensitivity to blockage. These characteristics of mmWave communications pose several challenges to fully exploiting the potential of mmWave communications, including integrated circuits and system design, interference management, spatial reuse, anti-blockage, and dynamics control [1]. Though mmWave has the disadvantages mentioned above, rich continuous and underutilized frequency resources are available in mmWave bands. As a result, mmWave has become a key enabler for future radio-communication technologies. The 3GPP defined frequency range designation (FR2) is 24.25–52.6 GHz, and corresponding operating bands are defined in Table 1.

With the digitalization of railways, there are more and more types of sensors installed on trains, and a large number of monitoring data are collected during train operation, which need to be transmitted back to the ground center for comprehensive analysis. The traditional method is for these data to be copies by staff after the train is put into the yard or depot, which has low working efficiency and poor real-time performance. With the wide spectrum resources of mmWave and the application of Massive MIMO, beamforming, spatial multiplexing, and other technologies, the data throughput of mmWave communication systems between train and trackside can reach more than Gbps [2] during the process of trains entering the depot or passing through hotspot areas, such as train stations or marshaling yards, at very low speeds, normally 5–10 km/h, so large amounts of monitoring data such as videos collected during train operation can be transmitted during this timespan to the ground via the mmWave communication system in a few seconds for further analysis. Thus, mmWave is a promising solution for this scenario as a supplementary means of 5G-R and can significantly improve the efficiency and real-time performance of train–ground data transmission.

### 1.1. Related Works

Scholars have carried out a series of related work with respect to mmWave. Corresponding research will be reviewed from four categories involving communication scenarios, frequency bands, and research methodology, as well as the object of the research.

From the communication scenario point of view, railway, metro, residential house, indoor commercial, tropical area, and many other scenarios are covered by the open literature. With respect to railway scenarios, in [3], channel characteristics are studied in the 5G mmWave band centered at 25.25 GHz for typical urban HSR scenarios, including straight and curved route shapes. In [4], the 28 GHz mmWave channel characteristics of rural railway scenarios are studied via a calibrated ray-tracing simulator. Ref. [5] also focuses on the 28 GHz mmWave channel characteristics of rural railway scenarios, in which the dominant propagation mechanisms (direct, penetration, reflection, scattering, etc.) are determined and the three-dimensional environment model and the electromagnetic parameters of different objects are calibrated. The automatic coupling application scenario is studied in [6], in which a dynamic train-to-train (T2T) mmWave propagation measurement campaign was conducted, and the received signal power in the open field area was analyzed and modeled using a two-ray path loss model. The results indicated that the received power next to a platform was higher compared to the open field counterpart due to strong contributions from the signals reflected by the platform. The work in [7] studied the wireless coverage for intra-wagon scenarios at the 60 GHz band; by considering the balance of the wireless coverage and the cost of the transmitters, the deployment of two transmitters in one wagon is suggested. Typical objects in the railway propagation environment are also important research areas. In [8], a significant analysis methodology, which analyzes and distinguishes the contributions of different railway objects to the mmWave propagation channel, is proposed. In [9], the influence of typical objects to the mmWave propagation channel is analyzed for “train-to-infrastructure” and “intra-wagon” railway scenarios with various configurations. In [10], a statistical mmWave channel modeling for railway communications backhaul in 5G networks was proposed, involving 28 GHz and 60 GHz, by simulating in a novel software, NYUSIM. Viaducts and tunnels are common scenarios of railway terrain, especially in high-speed railways. The work in [11] considered measurement-based ray-tracer calibration and channel analysis for high-speed railway viaduct scenarios at 93.2 GHz, and the conclusion that the typical structure of viaducts and the application of horn antenna lead to a small value of Rician K-Factor and RMS delay spread is drawn. In [12], the authors investigate the mmWave propagation characteristics of a high-speed moving train based on field measurements in tunnel and viaduct scenarios. The measurements were carried out at 28 GHza and path loss (PL) and other channel parameters, including delay spread and Doppler shift, were investigated. In [13], smart-rail mobility was discussed, and the authors identified the main technical challenges and the corresponding chances concerning the reference scenario modules, accurate and efficient simulation platforms, beamforming strategies, and handover design. A mmWave beamforming scheme for disaster detection in high-speed railways was proposed in [14]; the key point of the study was that the antenna array generates multiple beams with different beam widths in different frequency bands simultaneously and the multiple beams are responsible for different detection areas. Another way to improve the reliability of railway communication is interlaced redundant coverage; in [15], a location-fair-based mmWave stable beamforming scheme under interlaced redundant coverage architecture is proposed to improve the stability and reliability of high-speed railway communications. Hotspots, such as depots and shunting yards, are also important railway scenarios. The work in [16] presented a comprehensive study on millimeter-wave-based mobile hotspot network (MHN) systems for high-speed train communications, including system design, field trial, channel modeling based on measurement campaign, simulation, and validation. In [17], mmWave channel measurements are reported for a railway depot environment using a wideband channel sounder operating at 60 GHz; path loss, RMS delay, as well as K-factors were extracted. The vacuum tube ultra-high-speed train (UHST) is a hot topic worldwide, and is a potential development direction of future transportation. In [18], a three-dimensional non-stationary mmWave geometry-based stochastic model (GBSM) is proposed to investigate the channel characteristics of UHST channels in vacuum tube scenarios.

Concerning the frequency bands, 26 GHz, 28 GHz, 30 GHz, 38 GHz, 40 GHz, 60 GHz, 73 GHz, 90 GHz, 93.2 GHz, and up to 300 GHz have all been studied by scholars, among which, 28 GHz and 60 GHz seem to have a higher level of attention. In [19], the authors conducted a 28 GHz-band 5G experimental trial on the actual Shinkansen that runs at a maximum speed of 283 km/h, and the experimental trial could achieve throughput exceeding 1.0 Gbps and consecutive handovers among the three BSs. In [20], field experiments of a 28 GHz-band 5G system at indoor train station platforms were carried out, and through the actual measurements, the results confirmed that the prototype 5G system could achieve mobile broadband capacity (more than 1 Gbps), even when the UE was located anywhere at the indoor train station platform. Ref [4] focuses on channel characteristics in rural railway environments at 28 GHz, and the mmWave channel characteristics of rural railway scenarios are studied via a calibrated ray-tracing simulator. In [5], influence analysis of typical objects in rural railway environments at 28 GHz is performed. In [21], millimeter-wave (mmWave) high-speed train measurements were conducted at 28 GHz with a speed up to 170 km/h in two different HST scenarios, viaduct and tunnel, in which large- and small-scale fading characteristics were extracted.

Moving on to research methodology, measure campaigns using measurement results published in the open literature, ray-tracing simulations, ultrawideband (UWB) channel sounding, deep learning, deep reinforcement learning (DRL), machine learning(ML), the space alternating generalized expectation-maximization (SAGE) algorithm, simulation platforms such as MATLAB, NYUSIM, mathematical methods such as non-convex problem with a near-optimal solution, coalition game, as well as the Bayesian approach, have all been used by scholars. Among all these methodologies, measure campaigns are basically very costly and time-intensive [22]; thus, ray-tracing simulations calibrated with measured data are commonly used, which means only a relatively small number of channel samples can be obtained, and then measurements are thus used to quantify the accuracy of a ray tracer. In [9], propagation measurements are conducted in the mmWave band for the 12 most common railway materials, the corresponding electromagnetic parameters are obtained, and a 3D ray-tracing (RT) simulator is calibrated. In [23], a series of horizontal directional scan-sounding measurements are performed inside a real high-speed train wagon at 60 GHz and 300 GHz frequency bands, and the channel characteristics are extracted, based on which, a self-developed RT simulator is validated through the reconstruction of the three-dimensional wagon model and the calibration of the electromagnetic properties of the main materials. In [5], based on the channel measurement conducted at 28 GHz with train-to-infrastructure deployment in the rural railway environment, a ray tracing (RT) simulator is calibrated in terms of environment modeling and electromagnetic calculation. In [11], ray tracer (RT) is employed to simulate the propagation in a high-speed railway viaduct scenario at a frequency of 93.2 GHz. In the article, by comparing the path loss between simulation and measurement, the permittivity and loss tangent of concrete, which is the construction material of viaducts, are calibrated in the ray tracer. In [24], based on the wideband measurements conducted in the tunnel scenario by using the “mobile hotspot network” system, 3D ray tracing (RT) is calibrated and validated to explore more channel characteristics in different HSR scenarios. In [25], since the measurement of a mmWave channel is deficient for high mobility, the proposed channel model is developed using a ray-tracing (RT) simulator that is validated with the channel measurements performed in the HST scenario at 93.2 GHz. Ref. [26] focuses on the analysis of propagation characteristics for train–ground communication systems in tunnel scenarios at both low-frequency and mmWave bands, based on ray-tracing (RT) simulation, and the material parameters in the RT simulation are calibrated by measurement data collected in realistic tunnel environments.

As for the research object, except for channel parameters such as power delay profiles (PDP), path loss, Rician K-factor, root-mean-square (RMS) delay spread, azimuth spread of arrival, and azimuth spread of departure, reconfigurable intelligent surface (RIS), beamforming, Doppler shift, energy efficient, user association, resource allocation and computation offloading, the influence of meteorological attenuation, sensing, minimal distance between base stations, as well as mmWave network architectures for railway have all been involved. In [27], the authors proposed a RIS-assisted scheduling scheme for scheduling interrupt flows and improving quality of service (QoS), and in the proposed scheme, an RIS is deployed between the BS and multiple mobile relays (MRs). In [28], multiple beams with different beamwidths are formed by the base station (BS) simultaneously to improve the system capacity, and the mobile relays (MRs) are provided with the ability to adjust the receiver (RX) beams automatically to enhance the received signal-to-noise ratio (SNR). In [29], control information and user information are transmitted through the high-frequency micro base station (BS) and the low-frequency macro BS, respectively, and an optimized beam width and power allocation scheme is proposed, which is combined with mobile relay (MR) technology to utilize the architecture of large-scale antenna beamforming. In [30], dynamic and fixed beamforming is evaluated based on the generalized model and the measured data, and the results show that the average throughput of dynamic beamforming is only 4% higher than that of fixed beamforming in the HSR tunnel, but 21% higher in the train station when severe beam misalignment is present. In [31], the authors proposed a new Doppler shift estimator for mmWave communication systems on HSRs: an equally-divided structure-based estimator (ESBE) that divides the effective orthogonal frequency-division multiplexing (OFDM) symbol into multiple equal fragments. In [32], the modeling of the Doppler effect for mmWave in HSR communications is conducted, and data-aided Doppler estimation and compensation algorithms are designed based on the new model. In [33], the authors proposed a new machine learning-based Doppler shift estimator (MLDSE), which estimates the Doppler shift by using the reference signal received power (RSRP) values measured by the mobile receiver at all times.

### 1.2. Contribution and Organization of the Paper

Though a significant amount of research has been carried out from the four categories reviewed above, to the authors’ best knowledge, there is still a lack of evaluation of the 28 GHz mmWave channel characteristics for the railway marshaling yard scenario. However, accurate propagation modeling and parameter analysis are crucial for achieving better performance for the radio-communication system, from the perspective of both link level and system level. To fill the aforementioned research gaps, the novelty of this work is to investigate the railway marshaling yard mmWave propagation scenarios of 28 GHz mmWave and conduct a measurement campaign with the SDR channel sounding system, using self-developed software under the LabVIEW platform to derive channel parameters, based on which, the channel characteristic and influence of typical objects are analyzed. The data collection for this paper was conducted on a sunny morning, and meteorological effects, such as the potential impact of snow or rain, was not in the scope of this study.

The rest of this paper is structured as follows. In Section 2, we will describe the railway marshaling yard propagation scenario, and three testing scenarios are designed. The software-defined radio-based channel modeling system as well as construction of the measurement environment will be presented in Section 3. Then, corresponding analysis results are summarized in Section 4. Finally, Section 5 will present the conclusions.

## 2. Railway Marshaling Yard Propagation Scenario

We chose the railway marshaling yard located at the National Railway Track Test Center of China Academy Railway Sciences to conduct our measurement campaign. The marshaling yard, which has 8 tracks, is rectangle-shaped with a length and width of 600 m and 40 m, respectively, as shown in Figure 1.

Three typical propagation scenarios were designed in our measurement campaign, which, to the best knowledge of the authors, cover typical scenarios of the railway marshaling yard:

The first is an open field scene with Catenary poles distributed with a distance of 31 to 47 m on our testing track, and we chose the middle track as the testing track, as shown in Figure 2.

As shown in the red circle in Figure 2, the LOS path can be blocked by the horizontal metal beam or the metal net that is used to prevent birds from building nests at some testing points on the selected track; the influence will also be analyze in the following section.

The second is a representation of a locomotive on the adjacent track. To build this environment, a 26.6-m-long locomotive was placed on the adjacent track of our testing track, as shown in Figure 3.

The third is a representation of buildings near our testing track. To collect the data of this scenario, we chose the track next to the building, as shown in Figure 4.

The parameters of the corresponding scenarios are listed in Table 2.

## 3. Software-Defined Radio-Based Channel Modeling

### 3.1. SDR mmWave Measurement System

The SDR mmWave measurement system architecture is shown in Figure 5. The receiving and transmitting ends each use a USRP-2974 as the signal receiving and transmitting main body. Both USRP-2974s are connected to the GPS antenna on their side, so the internal GPS disciplined oscillator (GPSDO) can receive the GPS signal, which is used to lock the internal clock, so that the USRP-2974 at each end can be synchronized. By using a phased-array antenna at the receiving and transmitting ends, the intermediate frequency of 2.8 GHz is moved to the 28 GHz frequency band to achieve the transmission and reception of mmWave signals.

The compositions of the transmitting and receiving ends are shown in Figure 6.

At the transmitting end, the REF reference clock locked by the GPSDO will be sent to the RF module of the USRP-2974 and output to the up-converter of the phased-array antenna to ensure system synchronization. At the same time, the GPSDO will send the pulse-per-second signal (PPS) to the FPGA of the USRP-2974 for timing triggering. A ZC sequence is generated by the USRP-2974 host and sent to the USRP-2974 FPGA. After IQ compensation and digital up-conversion, the FPGA sends the digital baseband data to the USRP-2974 RF module, where DA conversion and baseband up-conversion are completed, and the signal is up-converted to 2.8 GHz. At the phased-array antenna end, the 2.8 GHz signal is received and mixed with the local oscillator, whose frequency is 25.2 GHz, to generate the final 28 GHz mmWave RF signal, which is transmitted through the antenna array surface. In this system, the transmitting angle of the phased-array antenna is 0 degrees, which is the normal direction of the array surface.

At the receiving end, the REF reference clock locked by the GPSDO will be sent to the RF module of the USRP-2974 and output to the down-converter of the phased-array antenna to ensure system synchronization. At the same time, the GPSDO will send PPS to the FPGA of the USRP-2974 for timing triggering. The antenna array will receive a 28 GHz mmWave signal, which will be mixed with the local oscillator, whose frequency is 25.2 GHz, and down convert it into a 2.8 GHz intermediate frequency signal. The RF module of the USRP-2974 will complete baseband down-conversion and baseband sampling and transmit the collected IQ data to the FPGA of the USRP-2974. After completing IQ compensation and digital down-conversion at the FPGA end, the data will be transmitted to the Host end. At the Host end, the same ZC waveform as the transmitter side will be generated and correlated with the received IQ data, and the correlated IQ CIR will be obtained. The IQ CIR will be converted from v to dBm to obtain the power delay profile (PDP); in the meantime, the original IQ waveform and PDP data will be recorded and stored on the disk.

#### 3.1.1. Key Hardware Components

The two most important components of the hardware of the measurement system are the receiver/transmitter USRP-2974 and the mmWave phased-array antenna mmPSA-TR64MX, as shown in Figure 7.

USRP-2974 is a high-performance software-defined radio (SDR) device produced by NI company, and is widely used for channel measurements in wireless communication systems. There are many advantages to using the USRP-2974 for channel measurements. Firstly, it provides high-performance analog-to-digital converters and processors that can achieve high-precision signal acquisition and analysis. Secondly, it supports a wide range of programming tool and development libraries, including GNU Radio, LabVIEW, and MATLAB, so that users can quickly customize their own channel measurement applications. The main parameters of the USRP-2974 are listed in Table 3.

The mmWave phased-array antenna mmPSA-TR64MX is a 64-unit two-dimensional mmWave phased-array antenna, the duplex mode of which is TDD, and the operating frequency band is 27–29 GHz. It supports the 3GPP 5G FR2 n257 frequency band and can be used for 5G mmWave wireless communication. The mmPSA-TR64MX is equipped with 64 phase-shifting transceiver channels, which can achieve high-performance dynamic beamforming of 4096 states in a two-dimensional ±$60^{\circ }$ degrees range, and the EIRP value is 51 dBm. The array can be individually turned on or off per 2 × 2 antenna unit to achieve flexible array reconstruction. The phased array is internally integrated with a bidirectional frequency conversion circuit and a high-performance mmWave local oscillator, realizing a 2–4 GHz S band IF transceiver interface, which is convenient for connection with the low-frequency band digital sampling system. The beam characteristics of the mmWave phased-array antenna are shown in Figure 8:

The main parameters of the mmWave phased-array antenna mmPSA-TR64MX are listed in Table 4.

#### 3.1.2. Self-Developed Sounding Data Process Software

The Laboratory Virtual Instrument Engineering Workbench (LabVIEW) platform was employed to develop our self-developed software in order to analyze the collected data. LabVIEW is a programming environment developed by NI company that uses graphical editing G language to write programs and is perfectly compatible with the USRP-2974 used in this system. The program developed by the authors for analyzing PDP and corresponding channel parameters based on collected data is shown in Figure 9.

### 3.2. Construction of Measurement Environment

The transmitting end of the measurement system was placed on the bridge over the railway tracks; the height of the bridge from the rail surface is 14 m and the transmitting antenna fixed on a tripod is 1.75 m high from the bridge, as shown in Figure 10.

The antenna of the receiving end is fixed on a tripod, which is placed at the middle of the selected track, and the receiving antenna is 1.6 m high from the rail surface, as shown in Figure 11.

In order to meet the data collection requirements described in Section 2, two tracks are selected for testing, as shown in Figure 12; the track marked with the red circle is selected for the first and second scenarios, and is denoted as $T_{1\&2}$. The track marked with the orange rectangle is selected for the third scenario, and is denoted as $T_{3}$. The data collection campaign was conduct on the corresponding track, with an interval of 5 m, for further analysis. On $T_{1\&2}$, data collection points with or without Catenary between the TX/RXs were separated into two categories for performance comparison.

### 3.3. Corresponding Algorithms

The IQ sampling rate used in the measurement system is 200 MS/s, and the corresponding PDP resolution is 1/200 M = 5 ns. The period of the ZC sequence is 10.24 µs.

In Figure 13, the blue blocks represent the sent ZC sequence and green blocks represent the received ZC sequence.

The process flow is shown in Figure 14.

#### 3.3.1. ZC Sequence Selection

In our measurement system, the sample rate of the ZC waveform is 200 MHz, the bandwidth is 160 MHz, and the length is 2048 points. The expression of the selected ZC sequence is:(1)$$x\left[n\right]=e^{\frac{j\pi un^{2}}{N}},0\leq n\leq N-1,N=2048$$
where *u* is the root index of the ZC sequence.

#### 3.3.2. Discrete Algorithm of Cross-Correlation

Define *h* as a discrete sequence, whose index can be negative. *N* is the number of elements in input sequence *X*, *M* is the number of elements in sequence *Y*, and assuming
(2)$$x_{j}=0,j<0\;or\;j\geq N$$
(3)$$y_{j}=0,j<0\;or\;j\geq M$$

*h* is calculated as:(4)$$h_{j}=\sum_{k=0}^{N-1}x_{k}\times y_{j+k}$$
where j∈[−(N−1),(M−1)].

The cross-correlation is calculated as:(5)$$Rxy_{i}=h_{i}-\left(N-1\right)$$
where i∈[0,N+M−2].

## 4. Results

As mentioned in Section 3.2, data samples are collected on the selected Tracks $T_{1\&2}$ and $T_{3}$ using the self-developed LabVIEW data processing software mentioned in Section 3.1.2. The channel sounding data are processed and the corresponding channel parameters, including received power and time delay, as well as the phase value of each multipath component, are extracted; at every position, 30 samples of each parameter are obtained, and the 95% statistical value are calculated for each parameter. The 95% statistical value of channel parameters at the position of 12.4 m (horizontal distance from the bridge) on $T_{1\&2}$ are shown in Table 5 and Figure 15; here, we show that the index of the MPC increases as the power decreases and the delay increases

Based on all the channel parameters derived from the collected sounding data, further analyses are made from the following two aspects.

### 4.1. Relationship between Parameters and Position

#### 4.1.1. Relationship between Number of MPCs and Position

As the index of the path increases, the quality of the sound data received from the path deteriorates, and the probability of extracting channel parameters from it decreases accordingly. If the number of decoded channel parameters exceeds 85% of the total decoding times for a specific path, then the path is determined as an effective path. The number of effective paths at each position is determined based on this principle, and the relationship between number of MPCs and position is shows in Figure 16.

As the distance between Tx and Rx increases, there is no significant change in the number of MPCs, which fluctuates randomly between 7 and 11; the mean value μ=9.55 and the variance δ=0.868.

#### 4.1.2. Relationship between Time-Spread and Position

For a specific sampling location, we denote the 95% statistical value of delay as $D_{first,95\%}$ for the first effective MPC and $D_{last,95\%}$ for the last effective MPC. The time-spread, which is denoted as TS for this location, is defined as:(6)$$TS=D_{last,95\%}-D_{first,95\%}$$

Using (Equation 6), the time-spread of every sampling position on $T_{1}$ is calculated, and the relationship between time-domain parameters and position is shown in Figure 17 and Figure 18.

Figure 17 shows the relationship between the delay of the first effective path and position. The nonlinear curve fitting method is used to fit the curve in Figure 17; the fitting curve is the red line in the figure, and the fitting equation is:(7)$$y=a\times x^{b}$$
where a=18.04322±3.86076 and b=0.74836±0.0436.

Figure 18 shows the relationship between time-spread and position; at the positions of 152.4 m and 202.4 m, there is a large time-spread sample in each of these two positions, which will be further analyzed in the section on the impact of typical objects. If we exclude these two samples for now, we get Figure 19.

The nonlinear curve fitting method is used to fit the curve in Figure 19; the fitting curve is the red line in the figure, and the fitting equation is:(8)$$y=\frac{a}{1+e^{\left(-k\times (x-xc)\right)}}$$
where a=2212.6593±778.51346, k=0.02413±0.0063 and xc=143.79056±36.49551.

#### 4.1.3. Relationship between Received Power and Position

The 95% statistical received power value of all effective paths in each position are calculated and are shown in Figure 20.

The 95% statistical power values are summed by position, and we get Figure 21.

The nonlinear curve fitting method is used to fit the curve in Figure 21; the fitting curve is the red line in the figure, and the fitting equation is:(9)y=b×ln(x−a)
where a=2.45198±3.83339 and b=−7.2857±0.15875. There is deep fading at the position of 152.2 m as well as 202.4 m, which will be further analyzed in the section on the impact of typical objects.

### 4.2. Impact of Typical Obstructions

#### 4.2.1. Impact of the Catenary

We use the parameters derived from the data collected on $T_{1\&2}$ to analyze the impact of the Catenary. As we mentioned in Section 2, the LOS path can be blocked by the horizontal metal beam or the metal net at some testing points on the selected track, as shown in Figure 2, and these testing points were recorded in detail during the test and are listed in Table 6.

From Figure 16 and Figure 17, we can tell that there are no obvious changes with respect to the number of MPCs and the delay of the first effective path at the blocked positions, and the LOS path, which is the first effective path, can penetrate the metal net and reach the receiving end, even though penetration loss is introduced, which we will further analyze. However, from Figure 18 and Figure 21, we can see that, in terms of the time-spread and received power, significant changes have taken place in these blocked positions.

In Figure 18, at the position of 37.4 m, we can see the time-spread does not change much. It is only increased by 118 ns compared with the expected value; however, at the positions of 152.4 m and 202.4 m, the time-spread compared with the expected value is increased by 2970 ns and 2853 ns, respectively. This is because, as the distance between Tx and Rx increases, the number of Catenary between the Tx and Rx increases. The delay of the first path, which is the LOS path, is basically stable, but the reflection and scattering conditions of other paths reaching the blocked position are more complex, so the delay spread increases obviously.

In Figure 21, at all the blocked positions, deep fading occurs; we can see at the positions of 37.4 m, 152.4 m, and 202.4 m that the penetration loss is 4.63 dB, 5.31 dB, and 10.33 dB, respectively. As mentioned above, as the distance between Tx and Rx increases, the number of Catenary between Tx and Rx increases, and the density of Catenary increases accordingly; therefore, the probability of being blocked by Catenary increases. We noticed that the penetration loss at the position of 202.4 m doubled compared with the former two positions; this is because, at this position, the path was blocked by Catenary twice, as we observed during the test. We can see that the penetration loss of the Catenary is approximately 5.07 dB.

#### 4.2.2. Impact of Adjacent Locomotive and Building

As we mentioned in Section 2, the locomotive, whose length is 26.6 m, was parked on the adjacent track of $T_{1}$ and covered the position from 52.4 m to 79 m.

From Figure 16, we can see the number of MPCs of positions near this area slightly increase by one to two paths. In Figure 18, we can see the time-spread of positions near this area slightly increases; this is because, with the introduction of the locomotive, the reflection and scattering conditions of paths become more complex. We can quantify the impact on the time-spread with MAPE:(10)$$MAPE=\frac{1}{n}\sum_{i=1}^{n}\left|\frac{Y_{i}-{\hat{Y}}_{i}}{Y_{i}}\right|\times 100\%$$

We select data from 42.4 m to 102.4 m, which are near the locomotive parking area, for fitting in order to obtain a higher fitting accuracy, and the actual time-spread data in the green circle in Figure 22 are used as well as the fitting data to calculate MAPE, as show in Figure 22.

The linear fitting equation is:(11)y=a+b×x
where a=84 and b=2.5. The actual data marked in the green circle are used to calculate MAPE, and we obtain MAPE=38.63%.

Now we analyze the impact of the building, which much larger than the locomotive, with a 7.5-m-high canopy and a 10-m-high main building, as shown in Figure 4. Since the locomotive and the building are basically set in parallel, we conduct the analysis through comparison of these two different objects, as shown in Figure 23.

In Figure 23a, the difference of MPC numbers derived from samples collected on $T_{1\&2}$ and $T_{3}$ is not obvious, with a deviation of only 1–2 paths randomly, and a maximum number of 12 MPCs on $T_{3}$.

In Figure 23b, the lines represent the 95% statistical delay value of the first effective path and indicate that the delay parameters derived from samples collected on $T_{1\&2}$ gradually increase as the distance increases, while the delay parameters derived from samples collected on $T_{3}$ exhibit an irregular fluctuation state as the distances increases.

In Figure 23c, the lines show the 95% statistical power values summed by position, and we see that the received power of $T_{1\&2}$ gradually decrease as the distance increases, while the delay of $T_{3}$ shows an irregular fluctuation state as the distances increases.

In Figure 23d, the lines show the time-spread of the first and the last effective path, and we see that the time-spread of $T_{1\&2}$ remains stable within a small range of 30 m, while the time-spread of $T_{3}$ shows an irregular fluctuation state as the distances increases.

Based on the above analysis, we can draw the conclusion that, due to the larger size and more irregular shape of the building, the received power, delay, and time-spread channel parameters within a small position range show dramatic random variations. Since normally there are several tracks in the railway marshaling yard, we recommend that locomotives with unloading data tasks travel on the tracks away from buildings, in order to obtain a more stable wireless transmission channel. In addition, considering the penetration loss caused by the Catenary, an additional signal-level protection margin needs to be added to achieve redundant protection and improve the reliability of the data transmission.

## 5. Conclusions

In this work, the railway marshaling yard mmWave propagation scenario is deeply analyzed and classified into three typical categories, involving open scene with Catenary poles, a locomotive parked on the adjacent track, and a building near the testing track.

A measurement campaign is conducted at the National Railway Track Test Center of CARS according to the three propagation scenarios, using an SDR channel sounding system, the hardware of which mainly consists of a USRP-2974 transceiver and a mmPSA-TR64MX phased-array antenna. A self-developed software under the LabVIEW platform is used to derive the channel parameters; to achieve this, a ZC sequence of 200 MHz sample rate and 2048 points is selected, and a discrete algorithm of cross-correlation is designed. Parameters involving received power, delay, and time-spread are successfully derived for every sampling position on our selected tracks of $T_{1\&2}$ and $T_{3}$, and the following four conclusions can be drawn based on these parameters:

(1) As the distance between Tx and Rx increase, there is no significant change in the number of MPCs, which fluctuates randomly between 7 and 11, with a mean value of μ=9.55 and variance of δ=0.868.

(2) The relationship between parameter and position is deeply analyzed and fitted with proper equations. The relationship between the delay of the first effective path and position conforms to the equation $y=a\times x^{b}$, the relationship between the time-spread and the position conforms to the equation $y=\frac{a}{1+e^{\left(-k\times (x-xc)\right)}}$, and the relation between received power and position conforms to the equation y=b×ln(x−a).

(3) The impact of the Catenary is an important factor. As the distance between Tx and Rx increases, the number of Catenary between the Tx and Rx increases, and the density of Catenary increases accordingly; therefore, the probability of being blocked by Catenary increases. This phenomenon affects mmWave propagation in two ways: With respect to received power, at all the blocked positions, deep fading occurs, with a penetration loss of 5.07 dB. As for time-spread, at the blocked position of 37.4 m, the time-spread does not change much, only increasing by 118 ns compared with the expected value; however, at the blocked positions of 152.4 m and 202.4 m, the time-spread compared with the expected value is increased by 2970 ns and 2853 ns, respectively, which means that as the distance between Tx and Rx increases, the time-spread increases accordingly at blocked positions.

(4) The impact of the locomotive parked on the adjacent track is not obvious; the time-spread slightly increases, with a MAPE of 38.63%. However, due to the larger size and more irregular shape of the building, the received power, delay, and time-spread channel parameters within a small position range show dramatic random variations, so the recommendation that locomotives with unloading data tasks travel on the tracks away from buildings is proposed.

The conclusions of this paper are helpful for facilitating the design of mmWave communication systems for railway marshaling yards, to fully exploit the potential of mmWave communications. In the meantime, 6G networks are expected to utilize ultra-high-frequencies such as mmWave in typical scenarios involving smart factory and cell-free communication, as well as non-terrestrial communication [34]. The relevant analysis methods employed in this paper can also be further extended and applied to research of mmWave utilization in 6G and future communication technologies for more scenarios. Moreover, meteorological effects are of great importance in the research of mmWave. In the future, we will conduct further data collection and analysis based on different weather conditions.

## Figures and Tables

**Figure 1 sensors-23-08108-f001:**
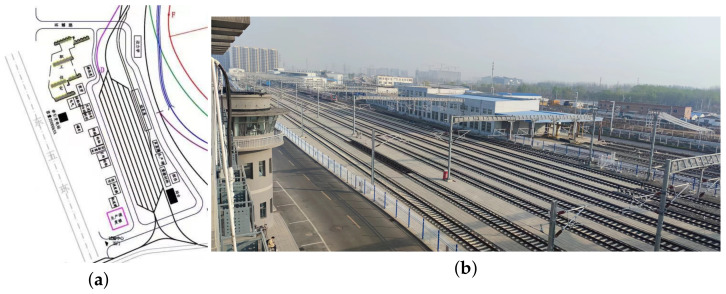
Railway marshaling yard. (**a**) Sketch map of the railway marshaling yard. (**b**) Picture of the railway marshaling yard.

**Figure 2 sensors-23-08108-f002:**
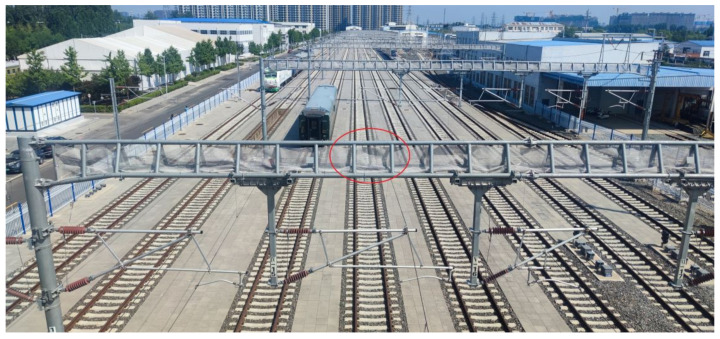
Open field scene with Catenary poles.

**Figure 3 sensors-23-08108-f003:**
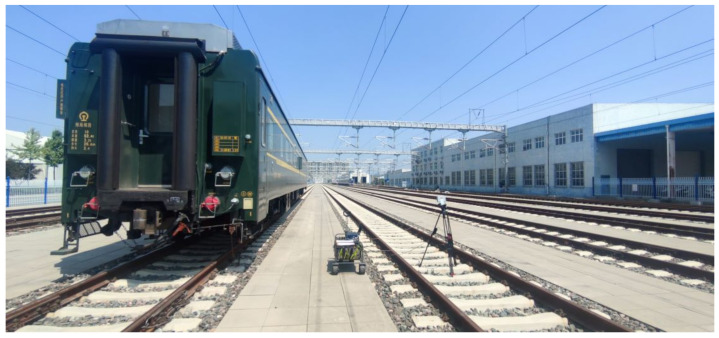
Locomotive on adjacent Track.

**Figure 4 sensors-23-08108-f004:**
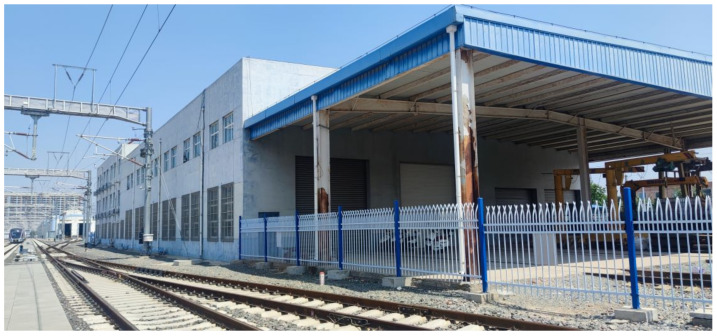
Building adjacent to the testing track.

**Figure 5 sensors-23-08108-f005:**
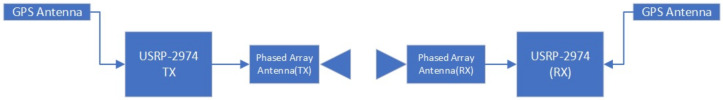
SDR mmWave measurement system architecture.

**Figure 6 sensors-23-08108-f006:**
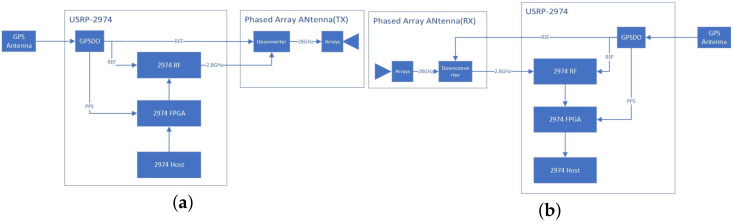
Composition diagram of transmitting and receiving ends. (**a**) Transmitting end. (**b**) Receiving end.

**Figure 7 sensors-23-08108-f007:**
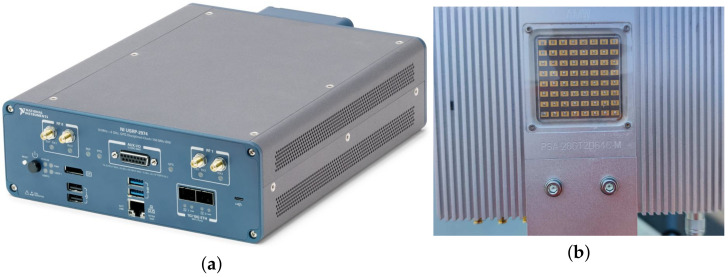
Receiver/transmitter and mmWave phased-array antenna. (**a**) USRP 2974. (**b**) mmWave phased-array antenna mmPSA-TR64MX.

**Figure 8 sensors-23-08108-f008:**
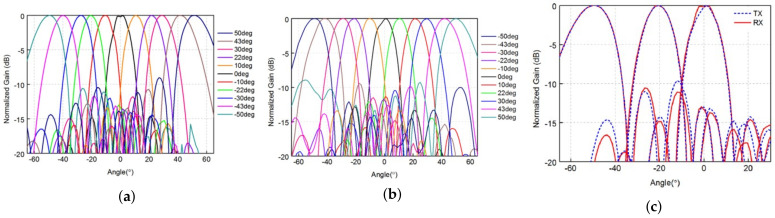
Beam Characteristics of the mmWave phased-array antenna. (**a**) Horizontal beam. (**b**) Vertical beam. (**c**) Beam Width.

**Figure 9 sensors-23-08108-f009:**
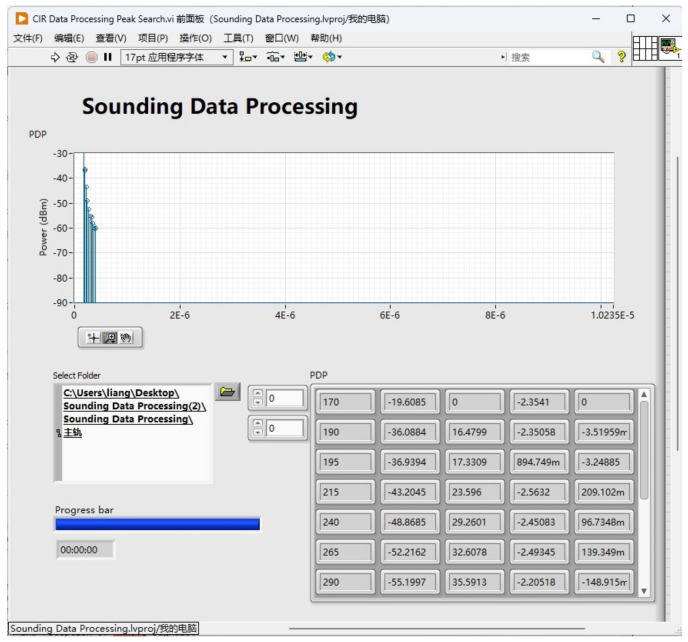
Self-developed software based on LabVIEW.

**Figure 10 sensors-23-08108-f010:**
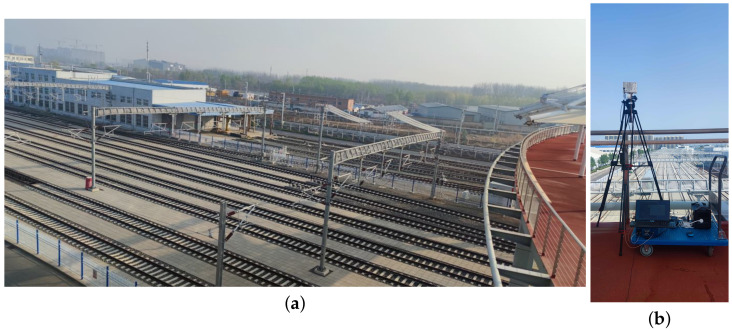
Construction of the transmitting end of the measurement system. (**a**) Bridge over the tracks. (**b**) Transmitting end on the bridge.

**Figure 11 sensors-23-08108-f011:**
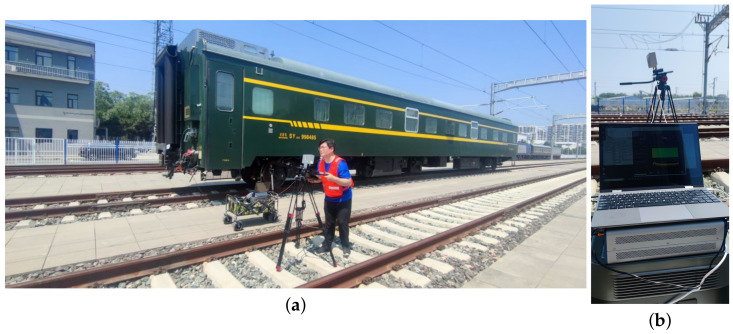
Construction of the receiving end of the measurement system. (**a**) Receiving antenna on the selected track. (**b**) Equipment of the receiving end.

**Figure 12 sensors-23-08108-f012:**
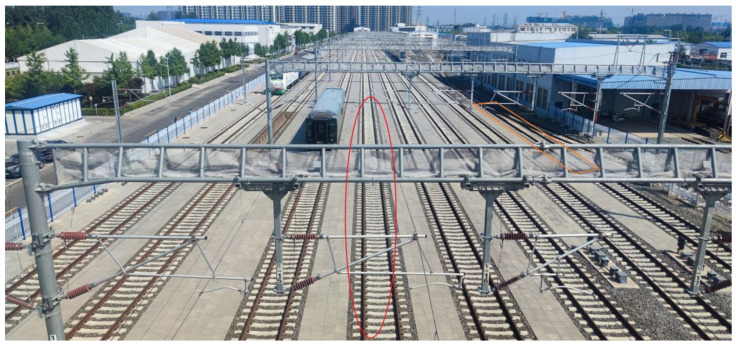
Selection of tracks.

**Figure 13 sensors-23-08108-f013:**
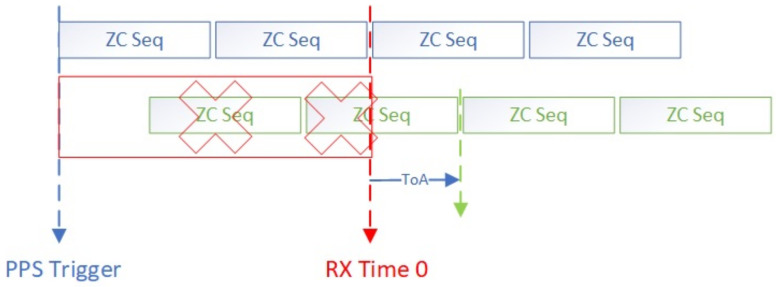
ZC sequence synchronization of the transmitting and receiving ends.

**Figure 14 sensors-23-08108-f014:**
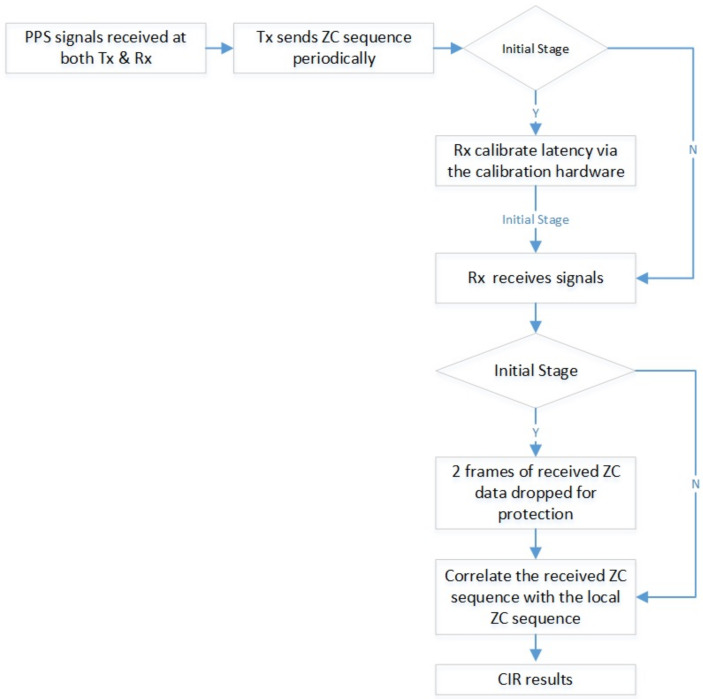
Process flow of the measurement system.

**Figure 15 sensors-23-08108-f015:**
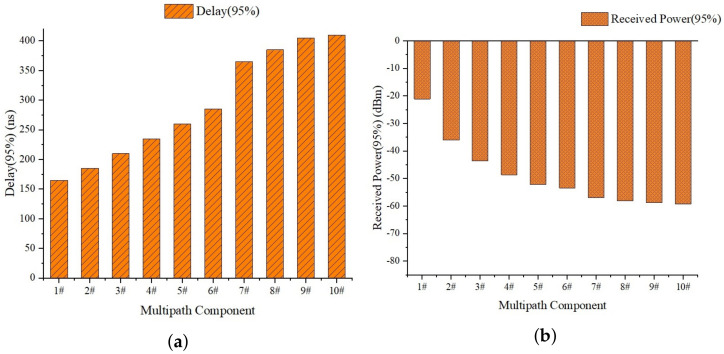
The channel parameters at the position of 12.4m. The horizontal axis coordinates 1# to 10# represent the index of MPC, which increases as the power decreases and the delay increases. (**a**) Delay. (**b**) Received Power.

**Figure 16 sensors-23-08108-f016:**
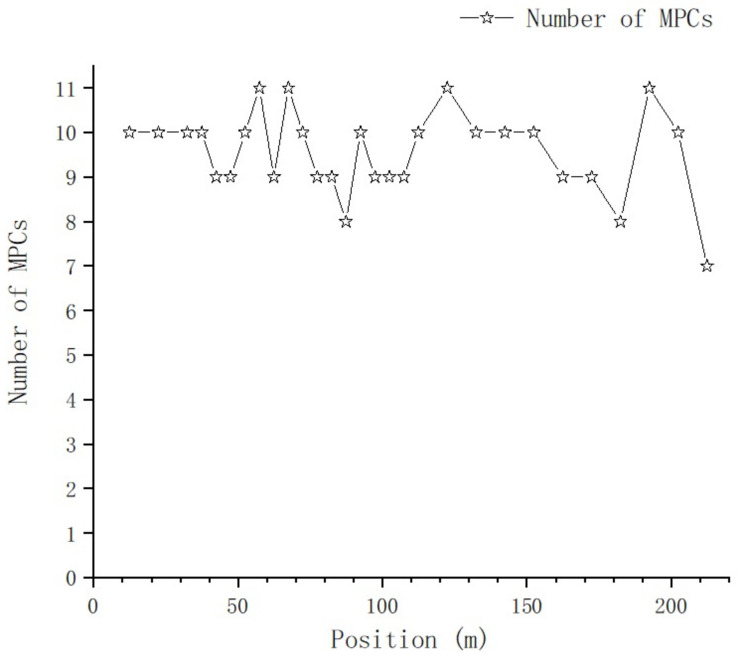
Relationship between number of MPCs and position.

**Figure 17 sensors-23-08108-f017:**
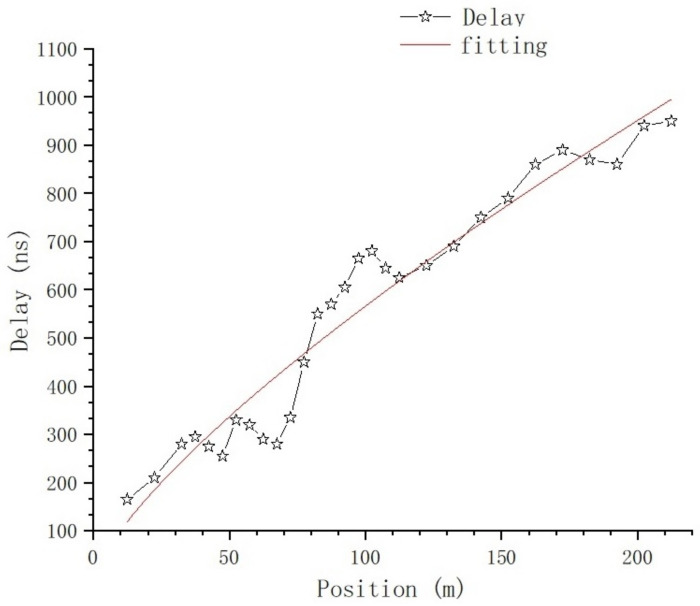
The relationship between delay of the first effective path and position.

**Figure 18 sensors-23-08108-f018:**
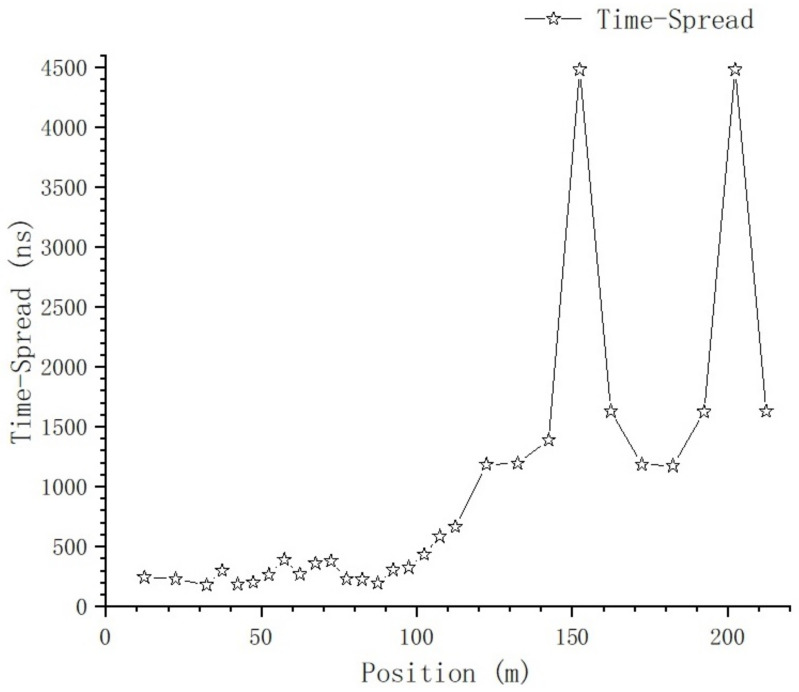
The relationship between time-spread and position.

**Figure 19 sensors-23-08108-f019:**
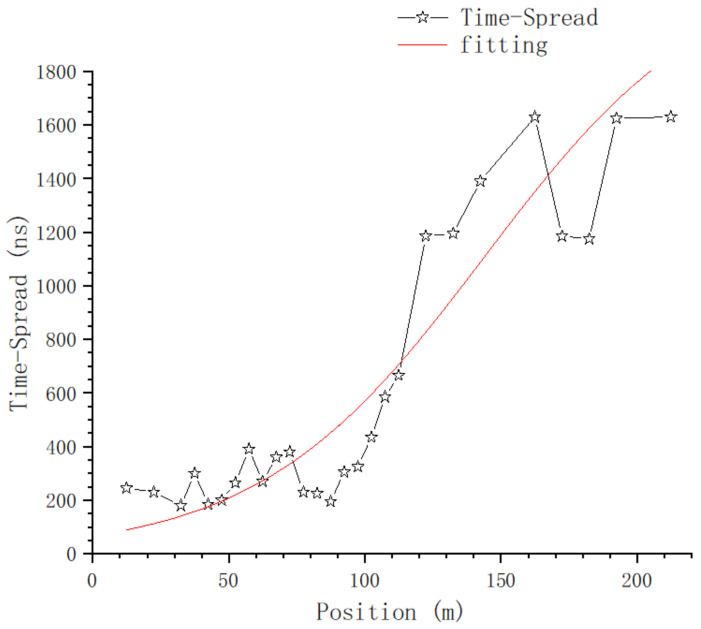
Fitting of time-spread.

**Figure 20 sensors-23-08108-f020:**
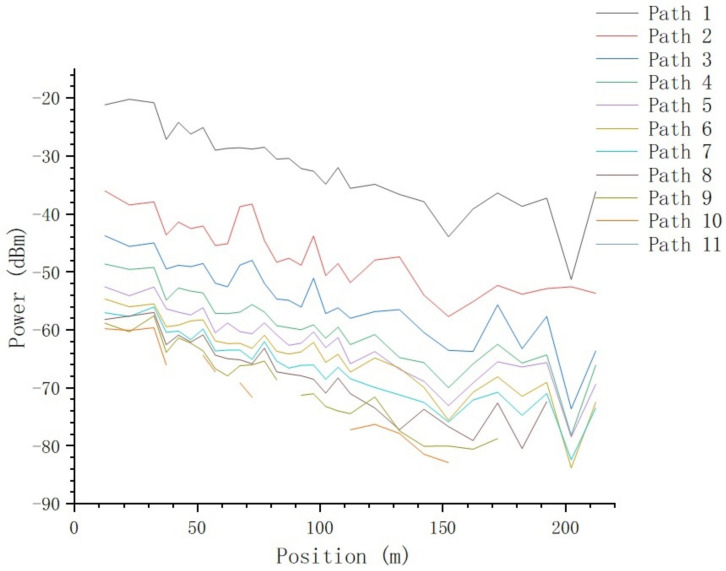
The relationship between received power and position.

**Figure 21 sensors-23-08108-f021:**
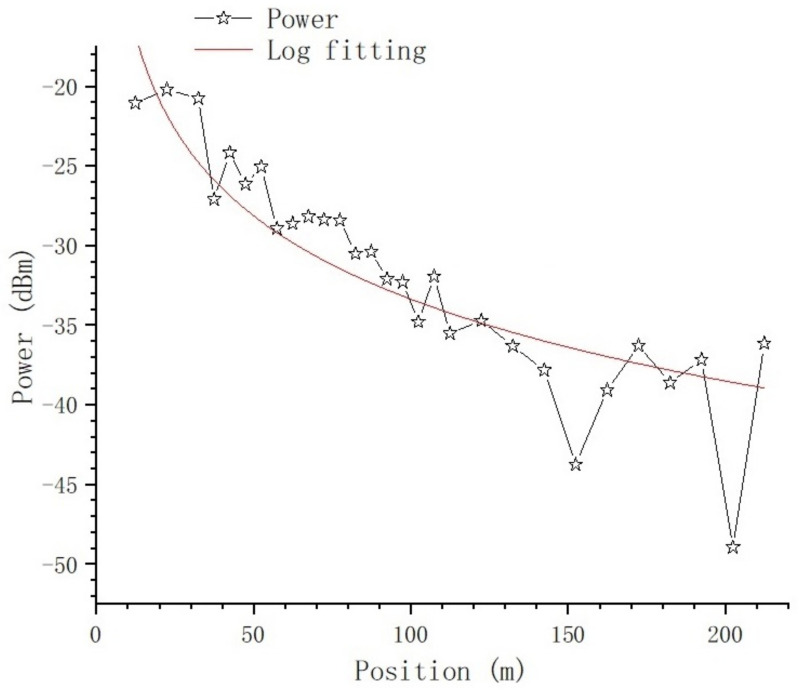
The relationship between summed received power and position.

**Figure 22 sensors-23-08108-f022:**
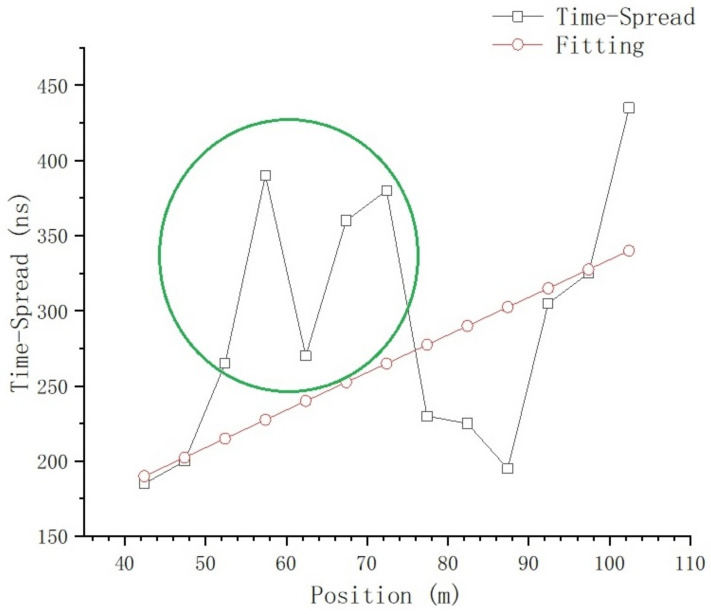
Fitting of time-spread from 42.4 m to 102.4 m.

**Figure 23 sensors-23-08108-f023:**
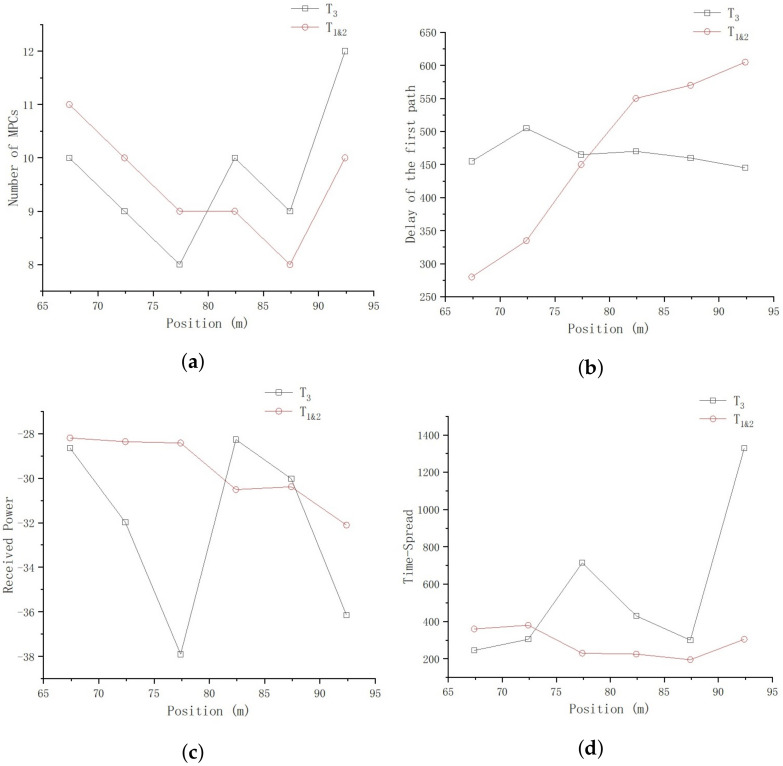
Comparison of the locomotive and the building. (**a**) Number of MPCs. (**b**) Received power. (**c**) Delay of the first effect path. (**d**) Time-spread.

**Table 1 sensors-23-08108-t001:** NR operating bands in FR2.

Operating Band	Uplink (GHz)	Downlink (GHz)	Duplex Mode
n257	26.5–29.5	26.5–29.5	TDD
n258	24.25–27.5	24.25–27.5	TDD
n260	37–40	37–40	TDD
n261	27.5–28.35	27.5–28.35	TDD

**Table 2 sensors-23-08108-t002:** Parameters of corresponding scenarios.

Parameter	Detail Information
Length of the yard	600 m
Width of the yard	40 m
Number of tracks	8
Track gauge	1.435 m
Distance between tracks	5.05 m
Distance between Catenary poles	31–47 m
Hight of Catenary poles	10 m
Hight of the building	canopy 7.5 m; building 10 m
Distance between the track and the building	6 m
Size of the locomotive	Length × Width × Height (m) = 26.6 × 3.105 × 4.433

**Table 3 sensors-23-08108-t003:** Parameters of USRP-2974.

Parameters		Detail Information
	ADC resolution	14 bit
	DAC resolution	16 bit
Baseband	Maximum I/Q sample rate	200 MS/s
	ADC sFDR	88 dB
	DAC sFDR	80 dB
	Number of channels	2
	Frequency range	10 MHz to 6 GHz
	Frequency step	<1 kHz
Transmitter	Maximum output power	5 mW to 100 mW
	Gain range	0 dB to 31.5 dB
	Gain step	0.5 dB
	Maximum instantaneous real-time bandwidth	160 MHz
	Number of channels	2
	Frequency range	10 MHz to 6 GHz
	Frequency step	<1 kHz
Receiver	Gain range	0 dB to 37.5 dB
	Gain step	0.5 dB
	Maximum input power	10 dBm
	Noise figure	5 dB to 7 dB
	Maximum instantaneous real-time bandwidth	160 MHz

**Table 4 sensors-23-08108-t004:** Parameters of the mmWave phased-array antenna mmPSA-TR64MX.

Parameters	Detail Information
Frequency Band	27–29 GHz
Duplex mode	TDD
Array size	8 × 8
Unit spacing	0.5 × λ
G/T	≥−7 dB/K
Polarization mode	Vertical polarization
Sidelobe suppression	≥15 dB @ normal-direction
Beam scanning range	±60° @ Horizontal&Vertical direction
Beam switching speed	<3 μs
Transceiver switching speed	<3 μs
EIRP	51 dBm
Gain dynamic range	≥30 dB
Gain adjustment accuracy	1 dB

**Table 5 sensors-23-08108-t005:** The channel parameters at the position of 12.4 m on $T_{1\&2}$.

Multipath Component	Delay (ns) (95%)	Received Power (dBm) (95%)
1	165	−21.15
2	185	−36.078
3	210	−43.687
4	235	−48.653
5	260	−52.216
6	285	−53.432
7	365	−56.995
8	385	−58.209
9	405	−58.767
10	410	−59.293

**Table 6 sensors-23-08108-t006:** Blocking positions on $T_{1}$.

Position (m)	Block Object
37.4	Middle of the medal net
42.4	Edge of the medal net
132.4	Edge of the medal net
152.4	Middle of the medal net
162.4	Edge of the medal net
202.4	Edge of the medal net
212.4	Middle of the medal net (twice)

## Data Availability

Data sharing not applicable.

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
