# Peer review of "SDR-Based 28 GHz mmWave Channel Modeling of Railway Marshaling Yard"

_sensors, 2023, doi:10.3390/s23198108_

Round 1
Reviewer 1 Report
The paper addresses the need for improved communication solutions in railway marshaling yards. It analyzes mmWave propagation in this specific context, conducts measurements, derives key channel parameters, and provides insights into how distance, obstructions, and position affect communication quality. The findings contribute to the development of effective mmWave systems for railway yards. The specific focus of this paper is on mmWave communication systems in a railway marshaling yard scenario. It seems that the is a need to evaluate such system performance the characteristics of the 28GHz mmWave channels. The scenario of the present paper has been categorized into three distinct categories: Open Scene with Catenary Poles: This likely refers to areas with overhead wires (catenary) used for powering trains, in a relatively open environment. Locomotive Parking on Adjacent Track: Investigates the impact of parked locomotives on communication quality on nearby tracks. Building Near the Testing Track: Focuses on the effect of buildings situated close to the testing track on communication. The study was carried out at the National Railway Track Test Center of CARS. Measurements were conducted based on the three propagation scenarios identified earlier. A Software-Defined Radio (SDR) channel sounding system was used for measurements. The hardware setup comprised of a transceiver USRP-2974 and a phased array antenna mmPSA-TR64MX. A self-developed software was used to extract key channel parameters from the collected data. The LabVIEW was utilized for developing this scenario. A discrete cross-correlation algorithm was designed to process the collected data and extract the required parameters. The key channel parameters successfully derived through this process include: Received Power: The power level of the signal received at the receiver. Delay: The time taken for the signal to travel from transmitter to receiver. Time-Spread: The dispersion of the signal in time due to multipath propagation. The analysis of the derived parameters led to the following four conclusions: The number of multipath components (MPCs) didn't exhibit significant change as the distance between transmitter and receiver increased. The mean value of MPCs was found to be approximately 9.55, with a variance of 0.868. The relationship between position and channel parameters was extensively studied and fitted with appropriate mathematical equations. The impact of the catenary system on signal penetration was investigated, leading to a proposed penetration loss of 5.07dB. Additionally, the time-spread increased when signal paths were blocked and Tx-Rx distance increased. The presence of parked locomotives near the track didn't have a major impact, but the irregular shapes of nearby buildings caused significant random variations in channel parameters within short positional ranges. Consequently, a recommendation was made to have locomotives with data tasks avoid tracks near buildings. The paper present an interesting idea. However, there are some comments to the author that need to be addressed before accepting this paper. The introduction is not very well organised. I would suggest to be divided into subsections. I would recommend to have the contribution separated subsection and the related works also be in a subsection and the paper results and findings would be in the last subsection. The text in the introduction line 119 needs to be modified. I would suggest to be the work in [7] studied The text in the introduction line 130 needs to be modified. I would suggest to be the work in [11] considered text in the introduction line 148 needs to be modified. I would suggest to be the work in [16] presented In line 207 space should be placed here profiles(PDP). Same comment to the line 276. Title in action 3 should be modified. Title in line 133 should be modified . The text from line 468 to line 473 needs to be modified. The author need to add reference about the importance of mmWave in further generation of wireless communication. In this regards I would suggest adding the following references [1 Niu, Y., Li, Y., Jin, D., Su, L. and Vasilakos, A.V., 2015. A survey of millimeter wave communications (mmWave) for 5G: opportunities and challenges. Wireless networks, 21, pp.2657-2676] [2] Alsabah, M., Naser, M.A., Mahmmod, B.M., Abdulhussain, S.H., Eissa, M.R., Al-Baidhani, A., Noordin, N.K., Sait, S.M., Al-Utaibi, K.A. and Hashim, F., 2021. 6G wireless communications networks: A comprehensive survey. Ieee Access, 9, pp.148191-148243.
The English of the paper needs t o be improved
Author Response
Please see the attachment.
Thank you very much for taking the time to review this manuscript.

Reviewer 2 Report
This paper proposes an SDR-based mmWave communication model, considering the dynamic railway environments. On-site experiments are conducted, and the results can be used for the design and performance evaluation of mmWave communication systems for railway marshaling yards.
Although this paper has made some contributions, it still requires major revisions.
(1)Mmwave can be affected by dynamic environmental factors. The authors consider the dynamic situation in railway environments. However, the work fails to address the meteorological effects. The authors should consider the potential impact of snow or rain on the mmWave signal and investigate whether these weather conditions could cause additional signal loss to the mmWave communication system.
(2)In section 4, the authors consider the impact of catenary on the mmWave communication. To make the conclusion more convincing, the author could include an experimental scenario with no catenary between the TX/RXs for performance comparison.
(3)The discussion between line 164 and line 167 is duplicated to the content between line 111 and line 114. Please revise this issue.
(4)Please revise the typo in line 463, "MATE" should be "MAPE".
(5) The abstract, introduction, and conclusion sections need to be refined and simplified. These sections are currently difficult for the readers to catch the main points.
Moderate editing of English language required.
Author Response

(The authors gave the same response as above.)

Reviewer 3 Report
This paper evaluates the 28GHz mmWave channel characteristics in railway marshalling yards, identifying three typical propagation scenarios and, based on derived parameters, recommends that locomotives with data tasks should travel on tracks away from large buildings due to significant variations in communication parameters. Due to the interest of the topic that it addresses, I find the work of utility for the scientific community. In this sense, I think that it could be suitable for publication in the Sensors journal provided that the following comments are implemented within the document:
- Both the abstract and conclusions sections seem to have overlapping content, which might be redundant. The same results and conclusions are reiterated multiple times. Streamline the content by ensuring that each piece of information is presented once, clearly, and concisely.
- The paper concludes with a recommendation concerning the position of the locomotive in relation to buildings. Expound on this recommendation by discussing its implications, potential challenges, and how it might be implemented.
- Every study or research has its limitations. Elaborate on the limitations of the current study. For instance, was the study limited to certain conditions, specific types of trains, or specific regions? Knowing the limitations can help other researchers understand the context better.
- Include a section discussing potential future work. Given the results of this study, what would be the next logical steps in this line of research? Are there other technologies or methodologies that could be explored in conjunction with the current findings?
- Given the findings of the paper, are there practical, real-world applications that can be immediately implemented or explored? Highlighting these can make the paper more impactful.
-
Author Response

(The authors gave the same response as above.)

Round 2
Reviewer 2 Report
The corresponding revisions have addressed all my concerns, this version could be accepted now.
The content is in good written.